# Light-evoked deformations in rod photoreceptors, pigment epithelium and subretinal space revealed by prolonged and multilayered optoretinography

Bingyao Tan[1,2,14], Huakun Li[3,14], Yueming Zhuo[4,5], Le Han[1,2], Rajeshkumar Mupparapu [1,2], Davide Nanni[3], Veluchamy Amutha Barathi[1,6,7], Daniel Palanker [4,8] ✉, Leopold Schmetterer[1,2,3,6,7,9,10,11,12] ✉ & Tong Ling [1,2,3,13] ✉

Phototransduction involves changes in concentration of ions and other solutes within photoreceptors and in subretinal space, which affect osmotic pressure and the associated water flow. Corresponding expansion and contraction of cellular layers can be imaged using optoretinography (ORG), based on phase-resolved optical coherence tomography (OCT). Until now, ORG could reliably detect only photoisomerization and phototransduction in photoreceptors, primarily in cones under bright stimuli. Here, by employing a phase-restoring subpixel motion correction algorithm, which enables imaging of the nanometer-scale tissue dynamics during minute-long recordings, and unsupervised learning of spatiotemporal patterns, we discover optical signatures of the other retinal structures' response to visual stimuli. These include inner and outer segments of rod photoreceptors, retinal pigment epithelium, and subretinal space in general. The high sensitivity of our technique enables detection of the retinal responses to dim stimuli: down to 0.01% bleach level, corresponding to natural levels of scotopic illumination. We also demonstrate that with a single flash, the optoretinogram can map retinal responses across a 12° field of view, potentially replacing multifocal electroretinography. This technique expands the diagnostic capabilities and practical applicability of optoretinography, providing an alternative to electroretinography, while combining structural and functional retinal imaging in the same OCT machine.

Phototransduction involves changes in concentration of ions and other solutes within photoreceptors and in subretinal space (SRS), which affect osmotic pressure and the associated water flow. Ion homeostasis of SRS is regulated by retinal pigment epithelium (RPE), a monolayer of cells between the photoreceptor outer segment and the choroid, and a variety of ocular diseases, such as age-related macular degeneration, retinitis pigmentosa, and Best's vitelliform macular dystrophy, are associated with dysfunction of epithelial transport of RPE cells[1–3]. In vivo assessment of RPE ion transport relies mostly on c-wave in electroretinography (ERG)[4] or electrooculography[5], which require contact electrodes and provide rather low spatial resolution.

Optoretinography (ORG) is an emerging imaging technology for non-invasive optical probing of retinal physiology in vivo. It usually utilizes phase-resolved optical coherence tomography (OCT) to detect

mechanical deformations of retinal cells in response to visual stimuli, for example, the deformation of photoreceptors' outer segments (OS)[6–11], which is associated with photoisomerization and phototransduction. Unlike conventional OCT, where its axial motion sensitivity is constrained by the bandwidth-limited axial resolution (typically >1 μm), axial sensitivity of the phase-resolved OCT is defined by the signal-to-noise ratio, potentially enabling the detection of nanometer-scale tissue dynamics[12]. ORG was recently employed to classify chromatic cone subtypes in human subjects based on their responses to flashes of various colors[13], and its clinical potential was demonstrated in assessing the progression of retinitis pigmentosa[14]. Furthermore, considering the similarity in photoreceptors' structure and function across mammals, small animals such as rodents, serve as convenient and cost-effective models for studying the mechanisms of photoreceptor degeneration using ORG[6,15]. In genetically modified animals, for example *Gnat1* and *Gnat2* mutants and knockouts, ORG can provide insights into the various stages of the phototransduction cascade[16,17]. In addition, unlike human patients, animals are usually free of other ocular pathologies, and their imaging under anesthesia experiences fewer motion artifacts.

Despite these potential benefits of conducting ORG studies in rodents, the axial resolution of near-infrared OCT (NIR-OCT) systems, typically limited to several micrometers in tissue, hinders the clear differentiation of rodent photoreceptor OS terminals and RPE, particularly given the interdigitation of microvilli with the OS. In a typical structural image captured by NIR-OCT, the OS, and RPE are compounded in a speckled layer (Fig. 1a). Moreover, since the backscattered light is coherently convolved with the point spread function of the optical system, a single pixel in the OCT image can contain a mixture of phase signals from multiple cells. As a result, separating the ORG signals from the OS and RPE presents a significant challenge.

In this paper, we present a robust, unsupervised learning approach to finding the hidden spatiotemporal patterns in the phase signals measured from speckles, which reveals optical signatures of the SRS, photoreceptor inner and outer segments, and RPE responses to light. Notably, image registration using our phase-restoring subpixel motion correction algorithm enables ORG recordings down to nanometer-scale for tens of seconds, as opposed to the typical few seconds in previous studies, allowing for detection of much slower and more subtle phenomena in retinal responses to visual stimuli. We demonstrate these light-evoked responses under various scotopic and photopic conditions and map the OS and SRS dynamics across a wide field using a single flash.

## Results

### Outer retina dynamics in response to visual stimuli

All ORG imaging experiments were performed in vivo using a custom-built NIR-OCT with an axial resolution of 2.0 μm in tissue. Cross-sectional (Fig. 1a) and volumetric scans were acquired in time sequences from a 12° field of view (FOV), corresponding to 0.8 mm on wild-type rat retinas[18]. A despeckled image, generated by averaging the neighboring B-scans from a rectangular region, was used to validate the delineation of retinal layers (Fig. 1b). As illustrated in Fig. 1, several hyperreflective layers were observed in the outer retina, including the external limiting membrane (ELM), the inner segment/outer segment junction (IS/OS), Bruch's membrane (BrM), and a thick speckling layer between the IS/OS and the BrM, which we call the mixed layer for convenience. As confirmed by the despeckled image, the mixed layer comprised photoreceptors' OS and RPE cells. To reveal the light-evoked dynamics in the outer retina, first we computed the changes in the optical path length (OPL) (phase difference) between the BrM and three hyperreflective bands in the outer retina, including ELM, IS/OS, and the top band of the mixed layer (the red color in the bottom right of Fig. 1a). To achieve high phase stability/sensitivity in vivo, one needs to correct the bulk tissue motion. For this purpose, we applied our recently developed phase-restoring motion correction method to register complex-valued OCT images with subpixel precision[19]. Regarding the uncorrected out-of-plane motion, our experiments demonstrated that even for one-minute-long recordings, the out-of-plane slow drift was typically below 2 μm (see Supplementary Discussion 1 and Supplementary Fig. 1)−much smaller than the beam diameter (12.2 μm, theoretical $1/e^2$ width, potentially even larger due to ocular aberrations). The phase uncertainty resulting from such uncorrected out-of-plane slow drift remained marginal, which allowed reliable measurements of retinal dynamics throughout prolonged ORG recordings (see Supplementary Discussion 1 and Supplementary Fig. 2). Phase traces extracted from the prolonged ORG experiments (55 s in duration) were spatially averaged across pixels within each

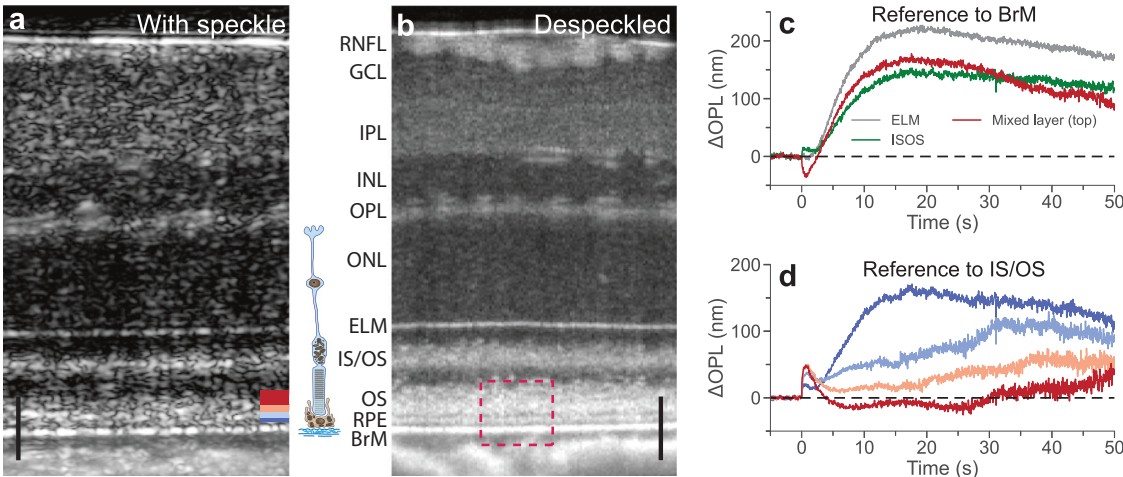

**Fig. 1 | Retinal layers and their dynamics in response to visual stimuli.**
**a** Averaged retinal B-scan (*n* = 125) from the same location. **b** Despeckled retinal B-scan, generated by averaging individual B-scans (*n* = 125) from a neighboring region, allows better resolving photoreceptors' OS, RPE, and BrM layers. An enlarged view of the magenta dashed box is shown in Fig. 3b. Both scans were acquired after 12 h of dark adaptation. Scale bar: 50 μm. **c** ORG signals obtained from various hyperreflective bands in the outer retina by taking the BrM as the reference. **d** ORG signals obtained at various depths in the mixed layer relative to IS/OS, with colors matching the corresponding bars at the bottom right of Fig. 1a. RNFL: retinal nerve fiber layer; GCL: ganglion cell layer; IPL: inner plexiform layer; INL: inner nuclear layer; OPL: outer plexiform layer; ONL: outer nuclear layer; ELM: external limiting membrane; IS/OS: inner segment/outer segment junction; OS: outer segment; RPE: retinal pigment epithelium; BrM: Bruch's membrane. Source data are provided as a Source Data file.

band. Note that in this study, the temporal phase change was extracted from a target layer with respect to a reference layer, which could be located either anterior or posterior to the target layer (see Supplementary Method 1). For consistency in interpretation of the increase/decrease of OPL as the expansion/contraction between the two layers, a negative sign was added when converting the phase change into the OPL change if the target layer was anterior to the reference layer.

As shown in Fig. 1c, distance between ELM and BrM increases after the stimulus (1 ms, wavelength 500 nm, 0.18% bleach level) at a rate of about 26 nm/s, reaching 220 nm around 20 s, and then slowly recovers afterwards. The IS/OS layer rapidly (-1 s) moves away from BrM, bounces back within 2 s and then continues to slowly move away from the BrM. The top red band of the mixed layer (presumably OS tips) rapidly (-1 s) moves toward the BrM by about 35 nm, then moves away from it over the next 20 s, followed by a slower recovery afterwards.

Conventionally, ORG monitors the movement of OS tips relative to the IS/OS layer. Due to the ambiguity of multiple tissue layers in NIR-OCT, we recorded the OPL changes from several bands in the mixed layer, indicated by various colors at the bottom right of Fig. 1a, relative to the IS/OS. As shown in Fig. 1d, the OPL changes in different bands contain a mixture of two distinct signatures: (a) rapid (-1 s) expansion followed by a slower (-10 s) decline, and (b) slow (-20 s) expansion, followed by even slower recovery. The superposition of distinct signals necessitates further signal decomposition and classification to determine the tissue origin associated with each individual signal.

## Unsupervised learning of spatiotemporal patterns for signal classification

The limited resolution of NIR-OCT leads to the mixing of light-evoked responses from both OS and RPE, thereby complicating the interpretation of the results. To identify distinct signal patterns in an unbiased manner and group them into distinct types, we projected individual phase traces onto a spatiotemporal feature space and employed unsupervised learning using agglomerative hierarchical clustering based on Ward's criterion[20]. We then trained a support vector machine (SVM) model on these type labels within the established feature space, enabling the extraction of each cluster's decision boundary and the classification of new phase traces using the same decision boundaries (Methods & Supplementary Fig. 3).

To account for inter-subject differences, we combined phase traces from five rats for feature extraction and subsequent unsupervised clustering (Supplementary Table 2 and Supplementary Table 3). We conducted preprocessing and principal component analysis (PCA) on the phase traces to extract their temporal features (Methods & Supplementary Fig. 4). We then constructed a three-dimensional (3D) spatiotemporal feature space using the distance to BrM (depth) as the spatial feature and the top two principal components (PCs) as the temporal features (Fig. 2a). Outliers in low-density regions were removed using a distance-based algorithm (gray dots in Fig. 2a).

To differentiate between two distinct signatures (Fig. 1d), we employed an agglomerative hierarchical clustering algorithm based on Ward's criterion[20]. By thresholding the dendrogram at the solid black line in Fig. 2b, the remaining phase traces (pink dots in Fig. 2a) were grouped into three clusters in the spatiotemporal feature space (Fig. 2c). A transition band (brown dots) facilitated better separation between the two distinct light-evoked tissue dynamics (Supplementary Fig. 5). Representative signals (Fig. 2d, red and blue dots in Fig. 2c) were obtained by averaging the individual phase traces within each cluster and converting them into OPL changes (ΔOPL). The first type of signal (red, Type-I) exhibited a rapid increase, peaking around 0.5 s, followed by a gradual decrease and a negative overshoot after 2.5 s (Fig. 2d). This Type-I signal matches the ORG signals from the photoreceptor outer segments in the literature, which were reported at shorter recording durations[8,10]. The second type of signal (blue, Type-

II) is characterized by a much slower rise, and its peak is not reached within the 3.5-second time range plotted in Fig. 2d.

Subsequently, we trained an SVM using the labels obtained from unsupervised clustering in Fig. 2c to set boundaries for Type-I (red surface) and Type-II (blue surface) signals in the spatiotemporal feature space (Fig. 3a). Using the pre-trained SVM, we successfully extracted Type-I and Type-II signals from new datasets. Type-I signals were located more anteriorly than Type-II signals, exhibiting a distribution close to normal (red line in Fig. 3b). Type-II signals were localized anterior to BrM, fitting a Gaussian function (blue curve in Fig. 3b). The locations of the peaks differed significantly ($*P < 0.05$, $n = 48$). The locations of Type-I and Type-II signals corresponded to the OS and RPE in the intensity profile, respectively, as determined by the despeckled image (yellow line in Fig. 3b). This observation suggested that the Type-I and Type-II signals correspond to the dynamics of the OS and RPE relative to IS/OS, respectively. We interpolated and clustered the phase traces from the prolonged recording using the SVM. The Type-I signal peaked within one second, underwent a negative undershoot peaking at 10 s, and returned to baseline within 30 s (Fig. 3c).

We further investigated the dynamics of the SRS, the extracellular space surrounding photoreceptors, and spanning from the apical membrane of RPE to the ELM[21]. The dynamics between IS/OS and ELM can be calculated by averaging the signals extracted from the pixels in the ELM layer, using the IS/OS layer as a reference. By summing up the Type-II signal (distance from RPE to IS/OS) and the OPL change between IS/OS and ELM, we obtained the SRS dynamics. As illustrated by the gray trace in Fig. 3c, the SRS signal increased more slowly, peaking at about 250 nm around 20 s, followed by an even slower recovery. In our subsequent quantitative analyses, we focus on OS and SRS dynamics.

## Signal dependence on stimulus parameters

We further quantified the signals in scotopic and photopic conditions by plotting the amplitude and latency of the peak of OS expansion and slope (expansion rate) from the SRS response (Fig. 4a). In scotopic conditions (Fig. 4b), the peak amplitude of the OS signal increased logarithmically with the stimulus strengths, from ΔOPL of about 10 (1.72) nm [mean (SD)] at a bleach level of 0.002% to about 53 (8.68) nm when the flash intensity increased 140-fold to 0.28%. The peak latencies remained within 440–500 ms at a bleach level <0.1%, but increased to 878 (140) ms and 1063 (108) ms in response to stimuli at 0.19% and 0.28% bleach level, respectively. In contrast, the slope of the SRS signal increased rapidly with the stimulus intensity up to 0.06% bleach level, and stabilized at the level of about 25 nm/s after that. In photopic conditions (Fig. 4c), the retina was pre-illuminated (500 nm) for 5 min and the flash was at 0.28% bleach level. Background illuminance below $6 \times 10^2$ photons/(μm² s) did not reduce the OS response, but it decreased 5-fold at background of $6 \times 10^4$ photons/(μm² s). The OS peak latency was not altered with an even stronger background - up to $6 \times 10^3$ photons/(μm² s), but it dropped by about 30% at $6 \times 10^4$ photons/(μm² s). Similarly, SRS expansion rate was affected by a background illuminance only above $6 \times 10^2$ photons/(μm² s), which reduced the slope by about 26% to 18.1 nm/s.

## *En-face* functional maps

Using repeated volumetric scans, we mapped the SRS and OS dynamics, analogous to multifocal ERG, but with a single flash. Figure 5a displays an OCT volume with a 12° FOV, with the structural contrast in gray and angiographic contrast highlighted in red. The spatiotemporal distribution of SRS and OS signals was obtained at a 0.10% bleach level. The temporal resolution was 8 Hz, and the total recording time was 5 s with 1 s baseline. Figure 5b shows the OS and SRS signals at specific time points. Figure 5c maps the spatial distribution of the phase traces, with each grid representing the average

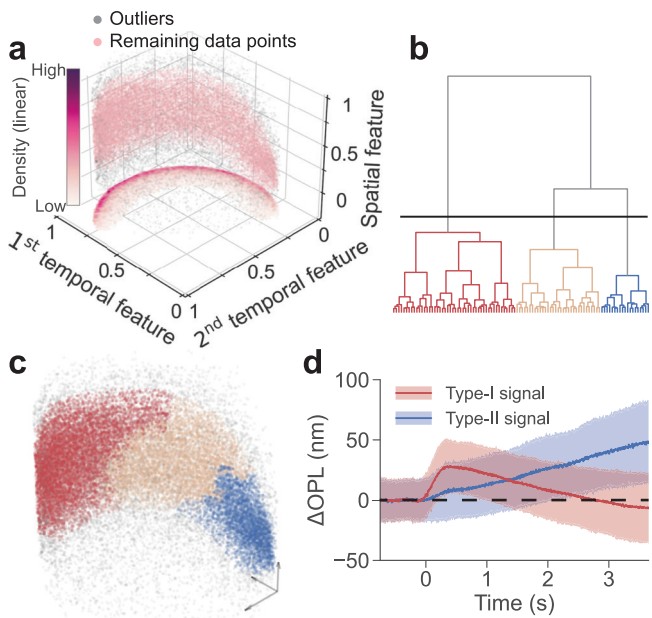

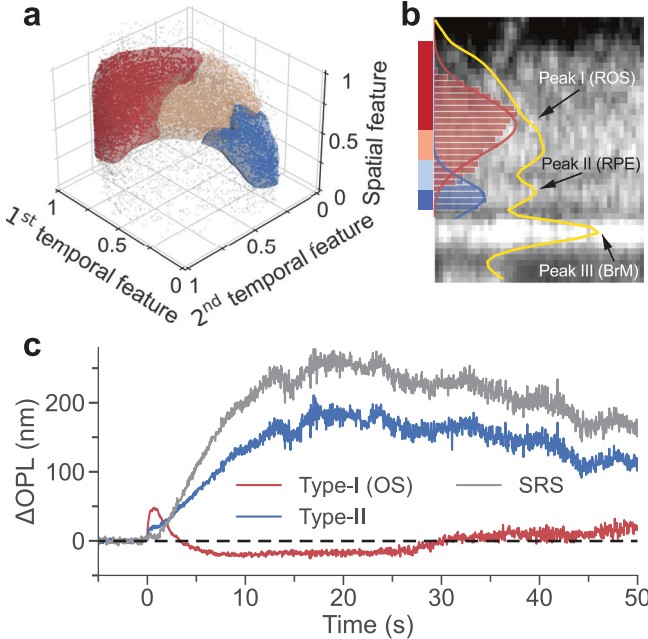

**Fig. 2 | Unsupervised clustering in the spatiotemporal feature space.**
**a** Distribution of the signals in the 3D spatiotemporal feature space. Gray dots represent the outliers identified using a distance-based detection method, while remaining data points were labeled in pink. The heatmap shows the distribution density of the remaining data points when projected onto the temporal feature plane. **b** Dendrogram showing the cluster structure of the remaining phase traces (pink dots in Fig. 2a) in the spatiotemporal feature space, with only the top 100 subclusters displayed. **c** Clustering the remaining phase traces in the spatio-temporal feature space into three clusters by thresholding the dendrogram along the solid black line. Colored dots denote different groups, and the gray dots denote outliers. **d** Corresponding representative Type-I and Type-II signals obtained by averaging the individual phase traces within each cluster. The solid lines and color bands denote the mean values and the range of standard deviations. Source data of Fig. 2d are provided as a Source Data file.

**Fig. 3 | Classification of new phase traces and validation of their origins. a** The trained SVM decision boundaries for the Type-I signal (red) and the Type-II signal (blue). The dots represent phase traces extracted from a new dataset, pre-processed, and projected onto the same 3D feature space. They were classified into Type-I signals (red dots), Type-II signals (blue dots), intermediate phase traces (brown dots), and outliers (gray dots). **b** An enlarged view of the dashed box in Fig. 1b, with contrast adjustment to enhance the RPE visibility. The histograms of Type-I (red bins) and Type-II (blue bins) signals, fitted by Gaussian functions (solid lines), display the depth distribution of the signals overlaid on top of the des-peckled structural image, with a yellow line representing the averaged intensity profile. The colored bars on the left correspond to the depth range from which the signals in Fig. 1d were extracted. **c** Type-I, Type-II, and SRS signals extracted from a prolonged recording. Source data of Fig. 3c are provided as a Source Data file.

response from a 1.2° × 0.48° area. In general, the OS and SRS signal maps show high-fidelity detection over the entire FOV, and high signal variance was observed underneath two large blood vessels, outlined by dashed lines in Fig. 5b, c. A video illustrating the spatiotemporal evolution of the OS and SRS signals can be found in Supplementary Movie 1.

Signals from SRS and OS, recorded over a wide field in response to a single flash, offer a convenient approach to diagnostic mapping, in contrast to the slow and low-resolution multifocal ERG. This technique expands the practical applicability of optoretinography to studies of not only photoreceptors but also the RPE's control of water dynamics in SRS in health and disease, thus providing an alternative to ERG.

## Discussion

After disambiguation of various signals in the outer retina, dynamics of its cellular layers can be described independently. Time derivative of the tissue deformation, i.e., its expansion rate, is very informative since it may reveal the water influx rate. As shown in Fig. 6, expansion rate of OS reaches its maximum of about 250 nm/s within 0.1 s after the stimulus and drops back to zero during the next 1 s. RPE, on the other hand, reaches the maximum contraction rate of about −10 nm/s at 2 s after the stimulus, leading to the maximum contraction of approximately 20 nm at 3 s, and slowly recovering afterwards. Expansion of SRS begins about 1.5 s after the stimulus, reaching a rate of 15 nm/s in the following 10 s, after which the expansion rate gradually decreases. SRS expansion may originate from water transport across the RPE in response to a decrease in $K^+$ and an increase in $Na^+$ concentrations due to phototransduction in photoreceptors, as observed earlier using

other methods[22,23]. The inner segment is compressed by about 10 nm during 1.5 s, after which it follows the dynamics of SRS, albeit at a lower amplitude (100 nm expansion at maximum).

It is interesting to relate the ORG signals to ERG previously recorded in the same species – pigmented rats, in response to a white flash[24], shown in Fig. 6e. The a-wave in ERG – the photoreceptors' response to light, begins right after the flash (within 10 ms), corresponding to the beginning of the OS expansion in ORG. Continuation of the a-wave is obscured by the b-wave, which starts a few tens of ms later and corresponds to the electric current generated by the second-order retinal neurons. C-wave in ERG takes over after about 0.5 s and reaches its maximum at around 1.5 s. It corresponds to a potassium current through RPE, induced by the photoreceptors' response to light. The RPE contraction in ORG, which reaches its maximum rate between 1 and 2 s, overlaps with the peak of c-wave (Fig. 6d, e). This optical signature opens the window into physiology of the photoreceptor-RPE interactions and the interphotoreceptor matrix. Water transport is limited by the membrane permeability and hence its dynamics is much slower than that of electric current. Using a model of the OS elongation due to osmotic imbalance during phototransduction[6], we estimated the membrane permeability coefficient based on the OS expansion rate. The water permeability coefficient of the rod OS membrane was found to be $5.9 \times 10^{-3}$ cm s$^{-1}$, not too far from $2.6 \times 10^{-3}$ cm s$^{-1}$ measured earlier in vitro – see Supplementary Discussion 2 for details.

Intriguingly, the hyperpolarized RPE layer is getting compressed in this process, unlike swelling of the hyperpolarized outer segments driven by osmotic pressure changes in response to released osmolytes during phototransduction.

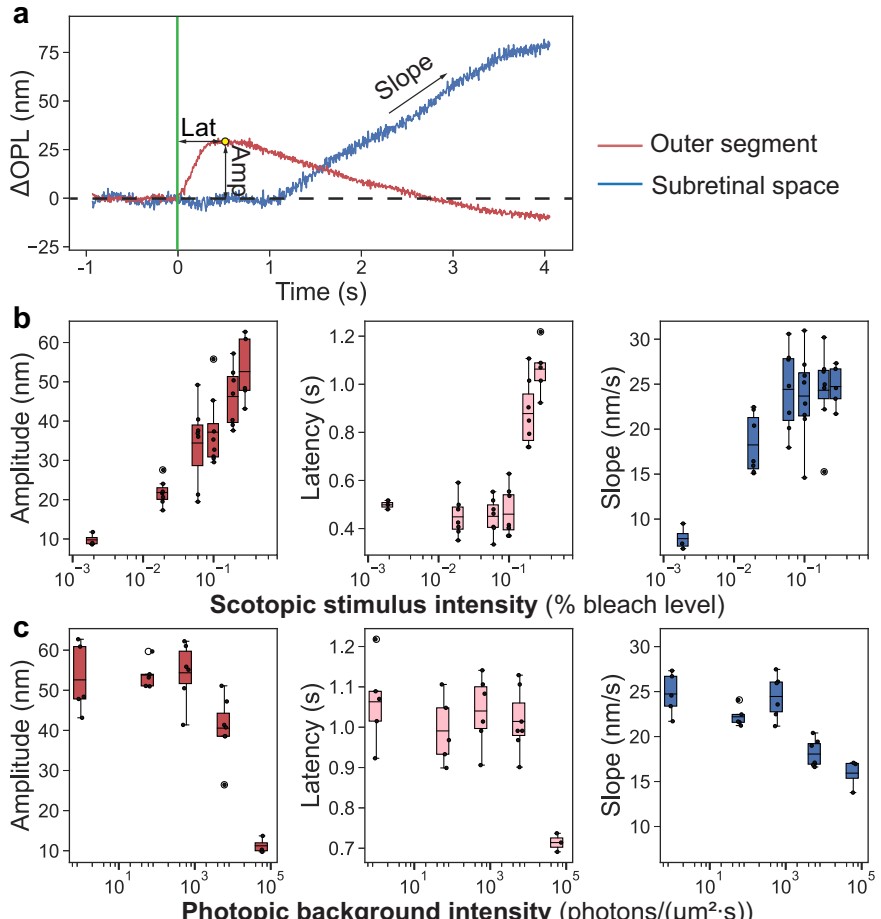

**Fig. 4 | Responses of the outer segment (OS) and subretinal space (SRS) in different conditions. a** Representative traces of the light-evoked responses in photoreceptor OS and in SRS. **b** Amplitude (Amp) and latency (Lat) of the OS response, and slope of the SRS expansion as a function of stimulus intensity on scotopic background ($n = 12$). **c** Same as a function of the background illuminance

($n = 15$). If the regeneration of rhodopsins is neglected, the 5-min background illumination at $6 \times 10^4$ photons/($\mu m^2$ s) would bleach 20.3% rhodopsins. For box plots, horizontal bar: mean value, box edges: 25 and 75 percentiles, whiskers: $1.5 \times$ standard deviations (SDs). Source data are provided as a Source Data file.

The responses of rods and cones to light vary in many aspects, including sensitivity, time constant, and light adaptation[25]. These differences may be attributed to different expressions of isoforms involved in the phototransduction cascade and distinct structural organizations of plasma membranes in rods and cones[26,27]. Until now, ORG was observed mainly in cone photoreceptors, since detecting rod signals in human subjects is more challenging because rods are generally smaller and densely packed around cones[28–30]. In rats, however, 97% of photoreceptors are rods[31], and hence our OS signals are likely dominated by the rod outer segments. Adaptive optics (AO) enables imaging of single rods in peripheral human retina, and one AO-OCT study reported that a flash bleaching 0.05% rhodopsin resulted in rod OS elongation of about 60 nm, while using the same flash bleaching 0.2% opsin did not result in a detectable cone OS elongation[10]. These results agree with our observations: 0.06% bleach caused OS elongation by 20-30 nm in rats.

Several studies investigated prolonged ORG signals using different methodologies. Lu et al. studied the changes in retinal layer thickness after strong visual stimuli (>23% rhodopsin bleach) by segmenting the layers from OCT structural images[32]. They examined micrometer-level responses during a 30-min dark adaptation process after the visual stimulus, with a temporal resolution of 2.1 s and a baseline signal fluctuation of hundreds of nanometers. Zhang et al. investigated the OCT intensity profile change in depths by averaging numerous A-scans. They observed a slow (peak latency of 10–100 s,

depending on the bleach level) increase in distance between IS/OS and BrM in response to light, and attributed it to the elongation of the OS[6]. Notably, this slow signal resembles our SRS expansion, while the actual OS signal in our observations and previous reports[10] rises and recovers much faster. A follow-up study by Pijewska et al. demonstrated that the phase-based ORG signals enabled higher detection sensitivity than the intensity-based processing methods[33]. Their results, obtained using strong stimuli (100% rhodopsin bleach), showed almost linear expansions between the ELM and BrM over 40 s after the visual stimuli were delivered.

Another interesting finding is the undershoot of the OS signal, with its tens of seconds-long recovery (Fig. 6a), which was not reported in previous studies due to shorter observation time. It may be related to water transport from the OS to SRS when the osmolytes ($G\alpha_t$, $G\beta_1\gamma_1$) rebind to the cell membrane during deactivation of phototransduction. Restoration of the OS osmotic pressure during expansion of the SRS may be accompanied by its slight (20 nm) compression, which recovers later along with the other cellular structures in the outer retina.

Expansion of the SRS, closely related to phototransduction and water transport via RPE, is larger than that of the OS. Such signature may serve as a more sensitive measure of retinal physiology, providing additional diagnostic insights for various diseases involving the outer retina. ORG conveniently combines structural and functional retinal imaging in the same machine.

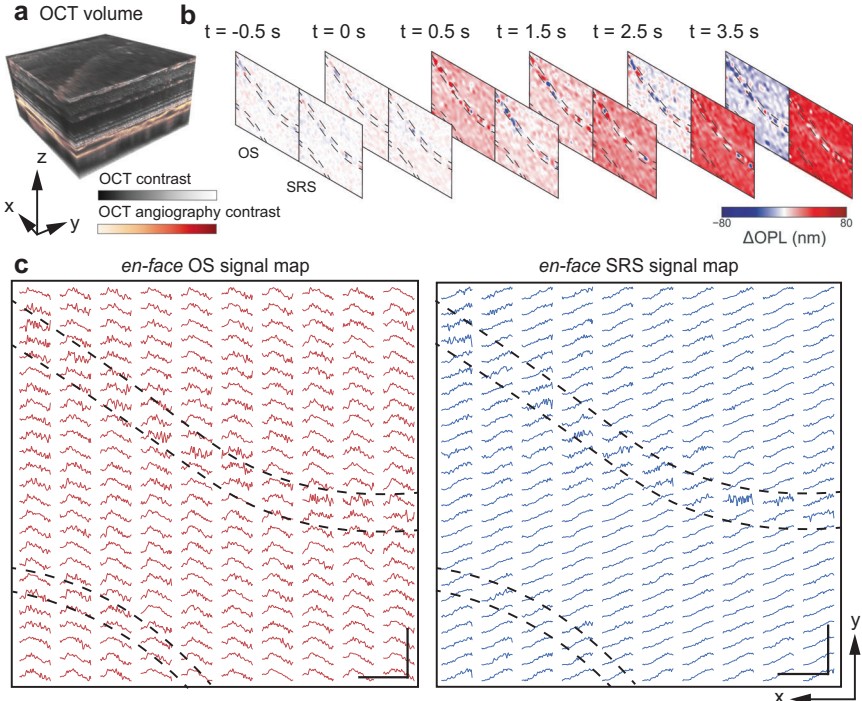

**Fig. 5 | Representative *en-face* maps of outer segment (OS) and subretinal space (SRS) signals. a** A volumetric scan covered a 12° field of view, with the structural contrast in gray and angiographic contrast highlighted in red. **b** The OS and SRS signals at selected time points. Locations blocked by large blood vessels are outlined by dashed lines. **c** Spatiotemporal evolution of the OS and SRS signals over the entire FOV. Each curve presents the average response from a 1.2° × 0.48° ($x \times y$) area. Scale bar: 100 µm. Source data of Fig. 5c are provided as a Source Data file.

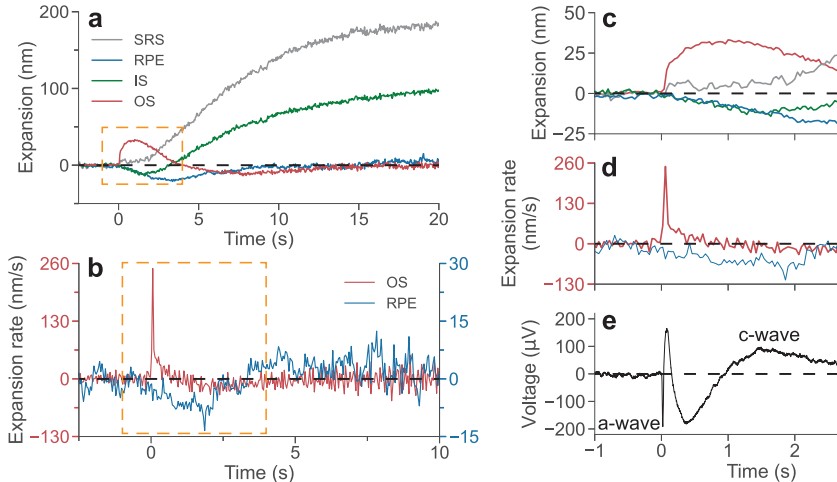

**Fig. 6 | Deformation of cellular layers over time and comparison of ORG with ERG. a** Representative dynamics of the deformations of the rod outer segment (OS), inner segment (IS), retinal pigment epithelium (RPE), and subretinal space (SRS) after a 1 ms green stimulus at 0.26% bleach level. **b** The expansion rate of the OS (red curve) and RPE (blue curve) averaged across 5 measurements. **c, d** Zoom-in view of the orange boxed data in (**a, b**), respectively. **e** An example electro-retinography (ERG) trace in response to a white flash, where the a-wave and c-wave are labeled. Modified from ref. 24. OPL change was converted into physical deformation with a refractive index of 1.41. Source data are provided as a Source Data file.

## Methods

### Ethical statement

All experiments were conducted in accordance with guidelines and approvals from Institutional Animal Care and Use Committee (IACUC), SingHealth (2020/SHS/1574).

### System setup

All the ORG imaging experiments in this study were performed using a custom-built spectral-domain OCT (Fig. 7a). This NIR-OCT system operated with a broadband superluminescent diode (cBLMD-T-850-HP-I, $\lambda_c = 840$ nm, $\Delta\lambda = 146$ nm, Superlum, Ireland), providing an axial resolution of 2.0 µm in tissue. A spectrometer interfaced with a line-scan camera (Cobra-800, Octoplus, E2V) acquired the spectral inter-ference fringes at an A-scan speed of 250 kHz, corresponding to an image depth of 1.07 mm in air. For posterior segment imaging in rodent eyes, a lens-based afocal telescope conjugated the axis of a galvo scanner to the pupil. To reduce the incident beam size and increase the scan angle, the telescope's magnification was 0.17 (scan

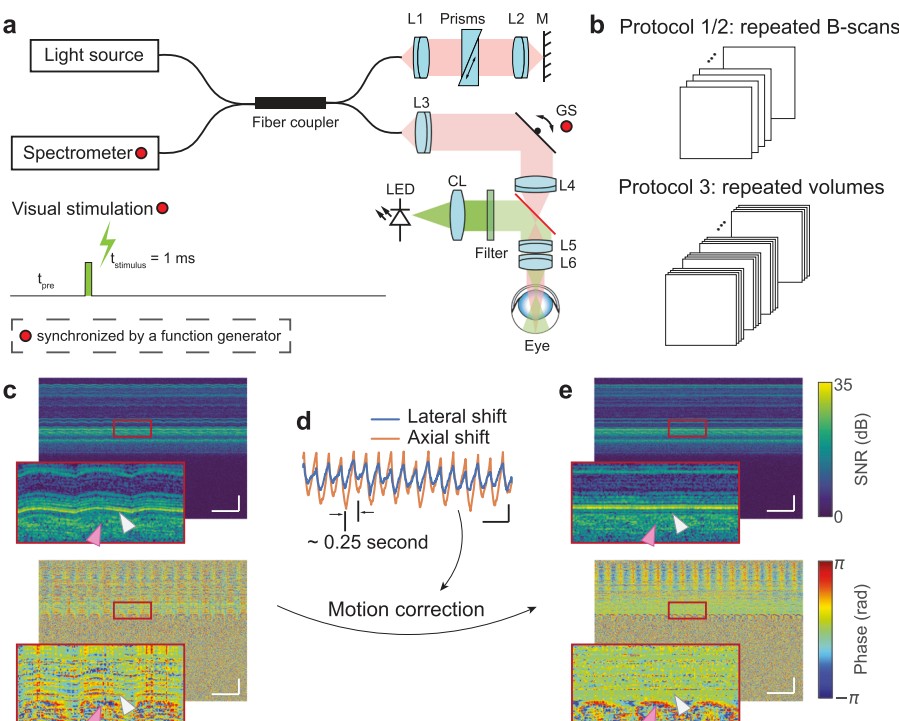

**Fig. 7 | System setup, stimulus scheme, acquisition protocols and image registration. a** A spectral-domain OCT was used to image the posterior segment of rat eyes. A line-scan camera interfaced with the spectrometer was used to acquire the interference fringes. A function generator synchronized the line-scan camera acquisition, galvo scanner rotation, and flash timing. L1-L6: doublet lenses. CL: condenser lens. GS: galvo scanner. Filter spectral window: 500 ± 5 nm. **b** Three acquisition protocols were used for ORG imaging: repeated B-scans (Protocol 1: 1000 A-scans per B-scan, 200 B-scans per second, Protocol 2: 1000 A-scans per B-scan, 25 B-scans per second) and repeated volumes (Protocol 3: 1000 A-scans per B-scan, 25 B-scans per volume, 8 volumes per second). The details of acquisition and stimulation protocols were listed in Supplementary Table 3. **c** Time-elapsed intensity (upper) and phase (lower) M-scans without correction, when no light stimulus was delivered to the retina. Scale bar: 500 ms (horizontal), 100 μm (vertical). **d** The subpixel-level bulk motion estimated by locating the peak of the upsampled cross-correlation map between repeated B-scans. Scale bar: 500 ms (horizontal), 2 μm (vertical). **e** Corresponding time-elapsed intensity (upper) and phase (lower) M-scans after the phase-restoring subpixel motion correction. Scale bar: 500 ms (horizontal), 100 μm (vertical). Gray and pink arrows in (**c**) and (**e**) label loci of the outer retina and the choroid, respectively. Source data of Fig. 7d are provided as a Source Data file.

lens: 80 mm focal length; ocular lens: 30 mm + 25 mm focal length). The theoretical diffraction-limited beam diameter was 12.2 μm (1/e$^2$ width) with a standard rat eye model[34]. A function generator (PCIe-6363, National Instruments, USA) synchronized the camera acquisition (both A-scan and B-scan acquisitions), galvanometer scanning, and visual stimulation. The data acquisition was controlled by an interface software developed using NI LabVIEW (19.0.1, National Instruments, TX, USA).

For visual stimulation, a LED (MCWHLP1, Thorlabs, USA) was collimated by an aspheric condenser lens, and a narrow bandpass filter (500 ± 5 nm, #65-694, Edmund Optics, Singapore) was used to reshape the spectrum to optimize the sensitivity to rhodopsin[35]. The LED's response time was ~300 μs, equivalent to 6% of the B-scan frame acquisition time, and LED's response waveform to trigger is shown in Supplementary Fig. 6. A 43.4° Maxwellian illumination was projected to the posterior eye to cover an area of approximately 6.75 mm$^2$. The power and duration of the light stimulus were converted to the bleach percentages of rhodopsin using a published method[36,37], detailed in Supplementary Method 2.

To generate a despeckled retinal image, volumetric raster scans (500 A-scans × 125 B-scans) were collected within a 12° × 0.3° (0.8 mm × 0.02 mm) rectangular FOV. The camera speed was set to 100 kHz to enhance the signal-to-noise ratio. Neighboring B-scans were aligned and then averaged into a single frame[38].

### Animal experiment protocol
Wild-type, Brown Norway rats (Rat Resource and Research Centre, male = 33, female = 15, age: 6–16 weeks) were used with details listed in Supplementary Table 2. Animals were sedated using a ketamine/xylazine combination to better maintain retinal functional responses compared to other commonly used anesthetics, such as isoflurane and urethane, while minimizing eye motion[39,40]. Vital signals, including heart and respiration rates, were monitored throughout the imaging sessions. Animals were placed in a prone position with their head restrained stereotaxically. Two mydriasis drops, 1% Tropicamide (Alcon, Geneva, Switzerland) and 2.5% Phenylephrine (Alcon, Geneva, Switzerland), were administrated onto the cornea before the imaging. The cornea was moisturized by a balanced salt solution frequently throughout the imaging sessions.

M-opsin and rhodopsin have similar sensitivity spectra (peaking at approximately 500 nm), while S-opsin's sensitivity peaks at 350 nm, with minimal overlap with rhodopsin sensitivity spectrum. Since M-opsin co-expression ratio decreases from ventral to dorsal[35], the scanning region was limited to the dorsal area to minimize the influence of M-opsin in cones.

Details of the acquisition and stimulation protocols for ORG imaging are listed in Supplementary Table 3. Three scanning protocols were used in this study. In the first protocol, we acquired 1000 A-scans per B-scan and 200 B-scans per second, with a total acquisition time of 5 s. In the second protocol, the B-scan time interval was increased to 40 ms, and 1375 B-scans were acquired over a period of 55 s. The third protocol consisted of 40 repeated volumes (25 B-scans per volume) recorded at 8 volumetric scans per second for a period of 5 s. Flash intensity, dark/light adaptation, and inter-flash time interval varied in different experiments.

## Automated extraction of the outer retina dynamics from speckle patterns

MATLAB (2020a, 2021a, and 2022b, MathWorks, MA, USA) was used for OCT image processing, registration, and all subsequent phase signal analysis. Python (v3.9.12) was used for plotting figures.

**Preprocessing of OCT images and phase traces.** The raw interference fringe first underwent standard OCT post-processing steps, including spectral calibration, k-linearity, dispersion compensation, and discrete Fourier transform (DFT), which resulted in complex-valued OCT images.

Phase-resolved OCT is very susceptible to bulk tissue motion, which results in apparent image distortions in the time-elapsed intensity M-scan and degrade the phase stability (see Fig. 7c). To correct the motion-induced phase error, we estimated subpixel-level translational displacements between repeated B-scans using the single-step DFT algorithm (see Fig. 7d)[41]. We registered the complex-valued OCT images using our recently developed phase-restoring subpixel motion correction algorithm[19], where the lateral and axial displacements were corrected by multiplying the corresponding exponential terms in the spatial frequency domain and the spectrum domain, respectively. After image registration (see Fig. 7e), we achieved excellent motion stability at photoreceptor layers (gray arrows), while periodical oscillations from the vascular pulsation (pink arrows) could be observed in the choroid.

Light-evoked dynamics of the outer retina was then extracted by computing the temporal phase difference between the pixel pairs from the outer retinal bands. Several hyperreflective bands, including ELM, IS/OS, BrM, and a thick speckling layer that contains photoreceptors' OS and RPE cells, were automatically segmented using graph theory and dynamic programming (Supplementary Fig. 3a)[42]. In self-referencing measurements, for each pixel in a target layer, we computed its temporal phase change with respect to a reference layer, detailed in Supplementary Method 1.

We also conducted temporal filtering to remove unwanted signal frequencies. The extracted signals were first processed by bandstop filters to minimize residual artifacts induced by heartbeat and breathing. A low-pass filter with a cut-off frequency of 10 Hz was subsequently used to filter out high-frequency oscillations. After filtering, both ends of the phase traces exhibited high variance due to edge effects, so they were removed from the data analysis afterwards (Supplementary Fig. 3c). For individual phase traces, our filters effectively captured the signal profiles while suppressing both periodic oscillations and high-frequency noise (Supplementary Fig. 3b). Meanwhile, it should be noted that the low-pass filter was implemented exclusively for the signal classification. After identifying pixels corresponding to the Type-I and Type-II signals, the light-evoked responses reported in this article were not processed by any low-pass filter to prevent potential signal distortion.

**Construction of spatiotemporal feature space.** We used the PCA to compress high-dimensional phase traces into a lower-dimensional feature space to facilitate the analysis (Supplementary Fig. 4c). Traces with a pre-stimulus SD larger than 60 mrad were excluded, mostly from pixels with low SNR or underneath blood vessels. Then each trace was normalized by subtracting its mean value and dividing by its SD (Supplementary Fig. 4). To account for inter-subject differences, we combined normalized traces extracted from five rats to calculate PC coefficients. The top two PCs capturing a total variance of 73.3% were selected as temporal features and its axial distance to the BrM was selected as the spatial feature. We normalized both the top two PCs and the depth information to construct a three-dimensional spatiotemporal feature space.

To avoid undesirable elongated clusters in subsequent unsupervised clustering analysis, we used a distance-based outlier detection method to preclude outliers distributed in low-density regions in the spatiotemporal feature space[43,44]. For each data point in the feature space, we calculated the minimum radius of a sphere that is centered at that point and can cover 2% of the remaining data points. This radius reflected the local distribution density around each data point in the feature space. Then, a given point would be labeled as an outlier if its corresponding radius was larger than $Q_3 + (Q_3 - Q_1)/5$, where $Q_1$ and $Q_3$ are the first and third quartiles of the calculated radii.

**Unsupervised clustering with a hierarchical clustering algorithm.** We used an agglomerative hierarchical clustering algorithm under the Ward criterion to group individual points in the established spatiotemporal feature space[20]. The Euclidean distance of each pair of points was computed and similar subclusters were then iteratively merged into larger clusters. Under the guidance of the Ward method or minimum variance method, each merger guaranteed a minimum increase of total within-cluster variance.

**Training a support vector model in the established feature space.** We trained a SVM in the spatiotemporal feature space with the previously obtained labels, including outlier, intermediate phase trace, Type-I signal, and Type-II signal, to obtain their decision boundaries. The input features were standardized before training, the Gaussian kernel was selected, and automatic hyperparameters optimization was turned on. Evaluated with the 10-fold cross-validation strategy, the trained SVM achieved a classification accuracy of 99.2%. The SVM is useful for processing a new dataset using the same criterion. Phase traces extracted from the new dataset will be preprocessed and projected onto the same feature space. Then Type-I and Type-II signals can be automatically extracted based on the classification results from the SVM.

**Processing of datasets with lower temporal resolution.** Prolonged recording (protocol 2) and volumetric scans (protocol 3) had an 8-fold and 25-fold lower temporal sampling rate than repeated B-scans (protocol 1), respectively. To cluster these phase traces in the same spatiotemporal feature space, they were first linearly interpolated into 5 ms time intervals and smoothed using a Gaussian filter. Other procedures were similar to those used for processing the repeated B-scans datasets.

### Statistics and reproducibility

Sample size calculation was not conducted for the pilot animal experiments. Comparison of depth between Type I and II signals was analyzed using paired, two-tailed Student's $t$ test, after confirming normality with the Shapiro-Wilk test. $p < 0.05$ was considered statistically significant. All statistical analyses were executed using MATLAB (R2020b, MathWorks).

Animals were randomly assigned to different study groups. Data with excessive motion artifacts (mainly due to heavier breathing effects of certain animals during anesthesia) were excluded from further analysis, because they introduced severe out-of-plane movement, which our image registration algorithm could not correct.

### Reporting summary

Further information on research design is available in the Nature Portfolio Reporting Summary linked to this article.

## Data availability

All data needed to evaluate the conclusions in the paper are present in the paper and the Supplementary Information. Two example datasets, one corresponding to a 5-second recording with an acquisition rate of 200 B-scans/second and the other corresponding to a prolonged recording with an acquisition rate of 25 B-scans/second, can be downloaded from https://github.com/NTU-Ling-lab/ORG-Classification. The raw experimental data are too large to be publicly shared, yet they are

available for research purposes from corresponding authors upon request. Requests will be fulfilled within 2 months.Source data are provided with this paper.

## Code availability

The algorithm demo for our prolonged and multilayered optoretinography (ORG) method can be found at https://github.com/NTU-Ling-lab/ORG-Classification[45].

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

## Acknowledgements

We thank Drs. R. Sabesan, A. Roorda, E. Pugh, and M. Ismail for fruitful discussions. This work was funded by grants from National Research Foundation Singapore (NRF-NRFF14-2022-0005, T.L.; NRF2019-THE002-0006, L.S.; NRF-CRP24-2020-0001, L.S.), the Startup Grant from Nanyang Technological University (T.L.), National Medical Research Council (NMRC/CG/C010A/2017, T.L.; CG/C010A/2017_SERI, L.S.; OFIRG/0048/2017, L.S.; OFLCG/004c/2018, L.S.; TA/MOH-000249-00/2018, L.S.; MOH-OFIRG20nov-0014, L.S.; NMRC/CG/M010/2017/Pre-Clinical, V.A.B.), the Ministry of Education, Singapore under its AcRF Tier 1 Grant (RS19/20, T.L.; RG28/21, T.L.), A*STAR (A20H4b0141, L.S.), the Singapore Eye Research Institute & Nanyang Technological University (SERI-NTU Advanced Ocular Engineering (STANCE) Program, L.S.), the Duke-NUS Medical School (Duke-NUS-KP(Coll)/2018/0009 A, L.S.), the SERI-Lee Foundation (LF1019-1, L.S.), NIH grant (U01 EY025501, D.P.) and AFOSR grant (FA9550-20-1-0186, D.P.).

## Author contributions

B.T., H.L., D.P., L.S., and T.L. designed the study. B.T. and L.S. built the NIR-OCT setup. B.T., H.L., D.N., and L.H. conducted the experiments. V.A.B. supported the animal preparation. B.T., H.L., R.M., L.H., and T.L. analyzed the data. B.T., H.L., Y.Z., D.P., L.S., and T.L. wrote the draft. All work was supervised by T.L., D.P., and L.S. T.L. and L.S. obtained funding. All authors contributed to the final manuscript.

## Competing interests

B.T., H.L., L.S., and T.L. are inventors on a PCT patent application (PCT/SG2024/050050) related to unsupervised signal classification and processing for optoretinography. The other authors declare no competing interests.

## Additional information

[1]Singapore Eye Research Institute, Singapore National Eye Centre, Singapore, Singapore. [2]SERI-NTU Advanced Ocular Engineering (STANCE) Program, Singapore, Singapore. [3]School of Chemistry, Chemical Engineering and Biotechnology, Nanyang Technological University, Singapore, Singapore. [4]Hansen Experimental Physics Laboratory, Stanford University, Stanford, CA 94305, USA. [5]Department of Electrical Engineering, Stanford University, Stanford, CA 94305, USA. [6]Department of Ophthalmology, Yong Loo Lin School of Medicine, National University of Singapore and National University Health System, Singapore, Singapore. [7]Ophthalmology and Visual Sciences Academic Clinical Program (Eye ACP), Duke-NUS Medical School, Singapore, Singapore. [8]Department of Ophthalmology, Stanford University, Stanford, CA 94305, USA. [9]Department of Ophthalmology, Lee Kong Chian School of Medicine, Nanyang Technological University, Singapore, Singapore. [10]Department of Clinical Pharmacology, Medical University of Vienna, Vienna, Austria. [11]Center for Medical Physics and Biomedical Engineering, Medical University of Vienna, Vienna, Austria. [12]Institute of Molecular and Clinical Ophthalmology, Basel, Switzerland. [13]School of Electrical and Electronic Engineering, Nanyang Technological University, Singapore, Singapore. [14]These authors contributed equally: Bingyao Tan, Huakun Li. ✉e-mail: palanker@stanford.edu; leopold.schmetterer@seri.com.sg; tong.ling@ntu.edu.sg

