## [Peer Review File · Nature Communications]

Reviewers' Comments:

Reviewer #1:

Remarks to the Author:

Review comments for manuscript NCOMMS-23-56338.

In this manuscript, the author reported a phase optoretinography study of wild-type rats. The author investigated the retinal response to light stimulation. Recorded the OCT image after the stimulation, exact and analyzed the phase change in different layers within around 1 minute. The author's study included different aspects, (1) the signal differences of different layers. (2) The signal was processed from repeated B-scans and enface images. (3) ORG comparison with ERG. In the analysis, the authors used unsupervised machine learning to differentiate the signal from different sources. The author demonstrated the capability of their custom-designed OCT system for multilayer ORG. The content delivery is good and the figures are nice. I have some comments and concerns that are listed below. I hope these comments can help help the author to improve the manuscript.

Major comments:

(1) Long-time ORG always suffers from the movement. For ORG computation, a general assumption is that the images are from the same location. However, for in vivo imaging, a static imaging condition is almost impossible. There is always movement from the imaging setups and the subject activities [1]. Although ORG for animals is much easier than for humans because we can put the animals in anesthesia. However, the movement related to the breath, heartbeat, and animal holder drift are all unignorable. Because of these factors, the long-time ORG for repeated OCT B-scan is very challenging. One efficient way to compensate for the movement is image registration, which is the way used in this study. Usually, this method is very powerful when the movement is within the image plane, however, this is not always the case, in a long recording, is very possible that the movement causes the scanning location to move out of the original location. If an OCT B-scan moves out of the image plane, caused by a movement, the 2D image registration would not work. An OCT B-scan can only cover a very small region, which is determined by the beam size, usually around 10 micrometers. Another aspect is that the phase ORG is very sensitive, it can detect the nanometer phase change. Thus, a long-time repeated OCT B-scan recording for ORG is very challenging.

In this manuscript, the author investigated ORG for about 1 minute. Although the authors claimed that they applied a very precise registration method, they did not justify if their system could maintain the image at the same location for such a long period. Without any information about this point, the following processing and computation are not based on a solid foundation.

(2) The author used machine learning for the signal and signal source analysis. It was a good attempt, however, it needs to be reconsidered. In this study, the author adopted unsupervised machine learning for the classification of two kinds of signals in the retinal band including OS tips and RPE cells. I have some questions and concerns: (1) In Figure 1A the vis-OCT B-scan we can see that the "mixed" band contains only the OS tip and the RPE layers. Why did the author intentionally separate it into four layers instead of two? This separation sequentially gave the complicated results in Figure 1D. Additionally, because separation into four, each band was very thin, which inevitably caused high fluctuation and segmentation errors. (2) In the spatial-temporal feature space, the author used two of the three clusters to represent the two types of ORG, how about the third type, it is just a transition of the other two types? Where is type III located in the "mixed" band? (3) From Figure 3B, we can see that the type I signal correlates with the OS tip, and the type II signal correlates with the RPE. Does it mean we can conclude that the signal type is mostly determined by the spatial information (depth)? If this is true, we get back to the point that the two kinds of signals are separated by the layer segmentation (depth difference).

Minor comments:

(1) The author may consider discussing the relationship between this study and other studies. Multilayer ORG and longtime monitoring [2], mouse phase ORG, also longtime monitoring [3].

(2) The author used the vis-OCT to show retinal layers that were not shown in the NIR-OCT B-scans. Later, the author also used the vis-OCT B-scan outer bands for the signal source investigation. In general, the vis-OCT has a better resolution, which should show more details than

the NIR-OCT. However, in this study, the authors need to pay attention to the fact that the retinal outer bands can be changed by dark or light adaptation conditions [4]. Normally the retina being imaged by vis-OCT should be light-adapted, and if the NIR-OCT image in the figure was from a dark-adapted retina, the comparison is not fair. In other words, to precisely guide the NIR-OCT outer band using a vis-OCT B-scan is risky, unless the adaptation condition is justified.

(3) In line 84, "a 12° field of view (FOV)", the author may consider adding the actual length like "xxx um" to help the reader to understand it better. We can see there is information in the material and method part, however, it is not very easy for the readers to find it.

(4) In Figure 1. For the NIR and vis-OCT B-scans, it seems like they are not from the same processing steps. The vis-OCT B-scan seems like is averaged by more frames. To show a better comparison, the author may consider processing the two images by the same method.

(5) In line 96, the citation is "bioRxiv (2022)", if it is officially published, please update the information.

(6) In line 97, the author said "Note that in this study, when converting the phase change into the OPL change, an additional negative sign was added if the target layer was anterior to the reference layer. This ensured that an increase in OPL always represented an expansion between the two layers." This sentence is confusing. This sentence is confusing and may lead to a possible interpretation that "OPL change" is always positive. the author may consider saying something like "The relative phase between two layers is calculated by subtracting the upper layer from the deeper layer, to ensure the relative phase is always positive".

(7) In line 166, the author explains the background illumination for the light adaptation. Can the author explain more about what the bleaching level these illumination backgrounds can cause?

Reference:

1. P. Zhang, E. B. Miller, S. K. Manna, R. K. Meleppat, E. N. Pugh Jr, and R. J. Zawadzki, "Temporal speckle-averaging of optical coherence tomography volumes for in-vivo cellular resolution neuronal and vascular retinal imaging," 6, 041105-041105 (2019).
2. C. D. Lu, B. Lee, J. Schottenhamml, A. Maier, E. N. Pugh, and J. G. Fujimoto, "Photoreceptor layer thickness changes during dark adaptation observed with ultrahigh-resolution optical coherence tomography," Investigative Ophthalmology Visual Science 58, 4632-4643 (2017).
3. E. Pijewska, P. Zhang, M. Meina, R. K. Meleppat, M. Szkulmowski, and R. J. Zawadzki, "Extraction of phase-based optoretinograms (ORG) from serial B-scans acquired over tens of seconds by mouse retinal raster scanning OCT system," 12, 7849-7871 (2021).
4. T.-H. Kim, J. Ding, and X. Yao, "Intrinsic signal optoretinography of dark adaptation kinetics," Scientific reports 12, 2475 (2022).

Reviewer #2:

Remarks to the Author:

The authors report novel findings from a carefully conducted, in-depth optical study to reveal in vivo nanometre-scale changes in the outer retina and retinal pigment epithelium following flash illumination. Some key advances over previous ORG studies include use of a novel motion correction algorithm to allow reliable tracking over the course of a minute and the characterisation of responses to very dim stimuli. Employment of unsupervised machine learning enabled distinction between two different types of response which had very different kinetics and which could be attributed to different retinal layers. They additionally present the effect of different stimulus and background parameters (intensities) on response kinetics. Importantly, the authors were also able to resolve and map responses across 12 degrees in a manner akin to multifocal electroretinography, but rapidly, with higher resolution and with a single stimulus. The study represents major technical and conceptual advances and would be an important addition to the literature, providing the basis for more refined in vivo investigation outer retinal physiology and pathophysiology.

Some comments are listed below.

Given that electroretinography permits direct (en masse) recording of neuronal electrical responses (which may not precisely correlate with optical/osmotic changes) including from inner

retinal neurons, I would not regard ORG as necessarily a replacement for electroretinography, but providing complementary information, likely with higher spatial resolution, particularly physiological data not directly obtainable with ERG.

Results

Line 101: "distance between ELM and BrM increases after the stimulus (1 ms, 500 nm, 0.18% bleach level)."

The 500 nm here is slightly confusing as to whether it refers to the distance or stimulus wavelength. I would suggest amending to, "distance between ELM and BrM increases after the stimulus (1 ms, wavelength 500 nm, 0.18% bleach level)."

Line 123: "we combined phase traces from five rats for feature extraction"

How were these 5 rats chosen? How many males/females? From the Methods, 43 rats were used in total with Table S2 giving the numbers in each test.

Line 153: "By incorporating the Type-II signal with the dynamics between IS/OS and ELM, we obtained the dynamics of SRS (from ELM to RPE)."

It was not clear to me precisely how the dynamics of the subretinal space were derived. Could the authors explain this more clearly?

Materials and Methods

Line 318: "A low-pass filter with a cut-off frequency of 10 Hz was subsequently used to filter out high-frequency oscillations". Given that much of phototransduction occurs within 100 ms, is there a possibility that physiologically meaningful data were lost/distorted by application of this filter? Did/can the authors check the effect of different cut-off frequencies and provide evidence or rationale to support this cut-off?

Reviewers' Comments:

Reviewer #1:

Remarks to the Author:

Review comments for manuscript NCOMMS-23-56338.

In this manuscript, the author reported a phase optoretinography study of wild-type rats. The author investigated the retinal response to light stimulation. Recorded the OCT image after the stimulation, exact and analyzed the phase change in different layers within around 1 minute. The author's study included different aspects, (1) the signal differences of different layers. (2) The signal was processed from repeated B-scans and enface images. (3) ORG comparison with ERG. In the analysis, the authors used unsupervised machine learning to differentiate the signal from different sources. The author demonstrated the capability of their custom-designed OCT system for multilayer ORG. The content delivery is good and the figures are nice. I have some comments and concerns that are listed below. I hope these comments can help help the author to improve the manuscript.

Major comments:

(1) Long-time ORG always suffers from the movement. For ORG computation, a general assumption is that the images are from the same location. However, for in vivo imaging, a static imaging condition is almost impossible. There is always movement from the imaging setups and the subject activities [1]. Although ORG for animals is much easier than for humans because we can put the animals in anesthesia. However, the movement related to the breath, heartbeat, and animal holder drift are all unignorable. Because of these factors, the long-time ORG for repeated OCT B-scan is very challenging. One efficient way to compensate for the movement is image registration, which is the way used in this study. Usually, this method is very powerful when the movement is within the image plane, however, this is not always the case, in a long recording, is very possible that the movement causes the scanning location to move out of the original location. If an OCT B-scan moves out of the image plane, caused by a movement, the 2D image registration would not work. An OCT B-scan can only cover a very small region, which is determined by the beam size, usually around 10 micrometers. Another aspect is that the phase ORG is very sensitive, it can detect the nanometer phase change. Thus, a long-time repeated OCT B-scan recording for ORG is very challenging.

In this manuscript, the author investigated ORG for about 1 minute. Although the authors claimed that they applied a very precise registration method, they did not justify if their system could maintain the image at the same location for such a long period. Without any information about this point, the following processing and computation are not based on a solid foundation.

(2) The author used machine learning for the signal and signal source analysis. It was a good attempt, however, it needs to be reconsidered. In this study, the author adopted unsupervised machine learning for the classification of two kinds of signals in the retinal band including OS tips and RPE cells. I have some questions and concerns: (1) In Figure 1A the vis-OCT B-scan we can see that the "mixed" band contains only the OS tip and the RPE layers. Why did the author intentionally separate it into four layers instead of two? This separation sequentially gave the complicated results in Figure 1D. Additionally, because separation into four, each band was very thin, which inevitably caused high fluctuation and segmentation errors. (2) In the spatial-temporal feature space, the author used two of the three clusters to represent the two types of ORG, how about the third type, it is just a transition of the other two types? Where is type III located in the "mixed" band? (3) From Figure 3B, we can see that the type I signal correlates with the OS tip, and the type II signal correlates with the RPE. Does it mean we can conclude that the signal type is mostly determined by the spatial information (depth)? If this is true, we get back to the point that the two kinds of signals are separated by the layer segmentation (depth difference).

Minor comments:

(1) The author may consider discussing the relationship between this study and other studies. Multilayer ORG and longtime monitoring [2], mouse phase ORG, also longtime monitoring [3].

(2) The author used the vis-OCT to show retinal layers that were not shown in the NIR-OCT B-scans. Later, the author also used the vis-OCT B-scan outer bands for the signal source investigation. In general, the vis-OCT has a better resolution, which should show more details than

the NIR-OCT. However, in this study, the authors need to pay attention to the fact that the retinal outer bands can be changed by dark or light adaptation conditions [4]. Normally the retina being imaged by vis-OCT should be light-adapted, and if the NIR-OCT image in the figure was from a dark-adapted retina, the comparison is not fair. In other words, to precisely guide the NIR-OCT outer band using a vis-OCT B-scan is risky, unless the adaptation condition is justified.

(3) In line 84, "a 12° field of view (FOV)", the author may consider adding the actual length like "xxx um" to help the reader to understand it better. We can see there is information in the material and method part, however, it is not very easy for the readers to find it.

(4) In Figure 1. For the NIR and vis-OCT B-scans, it seems like they are not from the same processing steps. The vis-OCT B-scan seems like is averaged by more frames. To show a better comparison, the author may consider processing the two images by the same method.

(5) In line 96, the citation is "bioRxiv (2022)", if it is officially published, please update the information.

(6) In line 97, the author said "Note that in this study, when converting the phase change into the OPL change, an additional negative sign was added if the target layer was anterior to the reference layer. This ensured that an increase in OPL always represented an expansion between the two layers." This sentence is confusing. This sentence is confusing and may lead to a possible interpretation that "OPL change" is always positive. the author may consider saying something like "The relative phase between two layers is calculated by subtracting the upper layer from the deeper layer, to ensure the relative phase is always positive".

(7) In line 166, the author explains the background illumination for the light adaptation. Can the author explain more about what the bleaching level these illumination backgrounds can cause?

Reference:

1. P. Zhang, E. B. Miller, S. K. Manna, R. K. Meleppat, E. N. Pugh Jr, and R. J. Zawadzki, "Temporal speckle-averaging of optical coherence tomography volumes for in-vivo cellular resolution neuronal and vascular retinal imaging," 6, 041105-041105 (2019).
2. C. D. Lu, B. Lee, J. Schottenhamml, A. Maier, E. N. Pugh, and J. G. Fujimoto, "Photoreceptor layer thickness changes during dark adaptation observed with ultrahigh-resolution optical coherence tomography," Investigative Ophthalmology Visual Science 58, 4632-4643 (2017).
3. E. Pijewska, P. Zhang, M. Meina, R. K. Meleppat, M. Szkulmowski, and R. J. Zawadzki, "Extraction of phase-based optoretinograms (ORG) from serial B-scans acquired over tens of seconds by mouse retinal raster scanning OCT system," 12, 7849-7871 (2021).
4. T.-H. Kim, J. Ding, and X. Yao, "Intrinsic signal optoretinography of dark adaptation kinetics," Scientific reports 12, 2475 (2022).

Reviewer #2:

Remarks to the Author:

The authors report novel findings from a carefully conducted, in-depth optical study to reveal in vivo nanometre-scale changes in the outer retina and retinal pigment epithelium following flash illumination. Some key advances over previous ORG studies include use of a novel motion correction algorithm to allow reliable tracking over the course of a minute and the characterisation of responses to very dim stimuli. Employment of unsupervised machine learning enabled distinction between two different types of response which had very different kinetics and which could be attributed to different retinal layers. They additionally present the effect of different stimulus and background parameters (intensities) on response kinetics. Importantly, the authors were also able to resolve and map responses across 12 degrees in a manner akin to multifocal electroretinography, but rapidly, with higher resolution and with a single stimulus. The study represents major technical and conceptual advances and would be an important addition to the literature, providing the basis for more refined in vivo investigation outer retinal physiology and pathophysiology.

Some comments are listed below.

Given that electroretinography permits direct (en masse) recording of neuronal electrical responses (which may not precisely correlate with optical/osmotic changes) including from inner

retinal neurons, I would not regard ORG as necessarily a replacement for electroretinography, but providing complementary information, likely with higher spatial resolution, particularly physiological data not directly obtainable with ERG.

Results

Line 101: "distance between ELM and BrM increases after the stimulus (1 ms, 500 nm, 0.18% bleach level)."

The 500 nm here is slightly confusing as to whether it refers to the distance or stimulus wavelength. I would suggest amending to, "distance between ELM and BrM increases after the stimulus (1 ms, wavelength 500 nm, 0.18% bleach level)."

Line 123: "we combined phase traces from five rats for feature extraction"

How were these 5 rats chosen? How many males/females? From the Methods, 43 rats were used in total with Table S2 giving the numbers in each test.

Line 153: "By incorporating the Type-II signal with the dynamics between IS/OS and ELM, we obtained the dynamics of SRS (from ELM to RPE)."

It was not clear to me precisely how the dynamics of the subretinal space were derived. Could the authors explain this more clearly?

Materials and Methods

Line 318: "A low-pass filter with a cut-off frequency of 10 Hz was subsequently used to filter out high-frequency oscillations". Given that much of phototransduction occurs within 100 ms, is there a possibility that physiologically meaningful data were lost/distorted by application of this filter? Did/can the authors check the effect of different cut-off frequencies and provide evidence or rationale to support this cut-off?

We would like to thank the reviewers and the editor for the careful reading of our manuscript, high praise for our results, and the detailed and constructive comments. We have revised the manuscript accordingly and conducted additional experiments for comments 1.1 and 1.6. Please find below our point-by-point responses (blue) and corresponding revisions to the manuscript (red).

Reviewer #1:

Major comments:

(1.1) Long-time ORG always suffers from the movement. For ORG computation, a general assumption is that the images are from the same location. However, for in vivo imaging, a static imaging condition is almost impossible. There is always movement from the imaging setups and the subject activities [1]. Although ORG for animals is much easier than for humans because we can put the animals in anesthesia. However, the movement related to the breath, heartbeat, and animal holder drift are all unignorable. Because of these factors, the long-time ORG for repeated OCT B-scan is very challenging. One efficient way to compensate for the movement is image registration, which is the way used in this study. Usually, this method is very powerful when the movement is within the image plane, however, this is not always the case, in a long recording, is very possible that the movement causes the scanning location to move out of the original location. If an OCT B-scan moves out of the image plane, caused by a movement, the 2D image registration would not work. An OCT B-scan can only cover a very small region, which is determined by the beam size, usually around 10 micrometers. Another aspect is that the phase ORG is very sensitive, it can detect the nanometer phase change. Thus, a long-time repeated OCT B-scan recording for ORG is very challenging.

In this manuscript, the author investigated ORG for about 1 minute. Although the authors claimed that they applied a very precise registration method, they did not justify if their system could maintain the image at the same location for such a long period. Without any information about this point, the following processing and computation are not based on a solid foundation.

Reference:

[1] P. Zhang, E. B. Miller, S. K. Manna, R. K. Meleppat, E. N. Pugh Jr, and R. J. Zawadzki, "Temporal speckle-averaging of optical coherence tomography volumes for in-vivo cellular resolution neuronal and vascular retinal imaging," 6, 041105-041105 (2019).

[Reply] We thank the reviewer for raising this question. Indeed, extracting nanoscopic phase-based ORG signals from a long-time repeated B-scan recording is not trivial, and we believe that the way we mount the animals by stereotaxic fixation and the precise subpixel motion correction we conducted, particularly in the axial direction, are the keys to achieving reproducible long-time recordings demonstrated in this study.

As the reviewer correctly pointed out, out-of-plane motion would remain uncorrected in the repeated B-scan mode. We have conducted new experiments to assess how significantly our measurements were affected by the out-of-plane motion. A cross-scanning pattern, consisting of a horizontal scan (following the lateral direction in the repeated B-scans mode) and a vertical scan

(following the out-of-plane direction in the repeated B-scans mode), was repeated over a duration of one minute. The temporal resolution was 45 ms, and the scanning areas in both directions were 12°. As shown in Supplementary Discussion Fig. 1, the rodents' eye movement in our experiments consists of periodic oscillations (< ± 2 μm) and a slow drift (typically below 2 μm with respect to the reference frame, Supplementary Discussion Fig. 1D), and their impacts on the phase uncertainty are discussed below:

1) The decorrelation noise associated with the periodic oscillations is repetitive and can be minimized by bandstop filters

In our previous study, we demonstrated that the decorrelation noise originating from a moving speckle pattern is a deterministic error rather than stochastic noise [Li *et al.*, 2022]. This suggests that the decorrelation noise, introduced by periodic oscillations at each pixel, exhibits a repetitive temporal pattern correlated with the oscillations, which can be directly filtered out if these oscillations stay within fixed frequency bands. Hence, we employed bandstop filters corresponding to the frequencies and harmonics of the breathing effect and heartbeat in the subsequent data processing (see Supplementary Fig. 1B), which mitigated the impact of decorrelation noise caused by the periodic oscillations in the out-of-plane direction.

2) The residual decorrelation noise caused by the slow drift in the out-of-plane direction remains insignificant, considering the relatively small displacement (<2 μm) with respect to our OCT beam size (12.2 μm, theoretical 1/e² diameter).

Since the three-dimensional point spread function (3D PSF) of the OCT imaging is compressed axially (2.1 μm FWHM in the axial direction; 7.2 μm FWHM in the lateral dimensions, potentially even larger due to ocular aberrations), our method has more tolerance to lateral eye movements. According to previous studies on OCT velocimetry [Vakoc *et al.*, 2009; Grafe *et al.*, 2019], if a sample undergoes a translational movement of ($\delta x, \delta y, \delta z$) between two measurements, the associated decorrelation noise (ϕ) follows a specific probability density function of,

$$P(\phi) = \frac{1-\alpha^2}{2\pi(1-\alpha^2 \cos^2 \phi)} \left[1 + \frac{\alpha \cos \phi}{\sqrt{1-\alpha^2 \cos^2 \phi}} (\pi - \cos^{-1}[\alpha \cos \phi]) \right],$$

$$\alpha = \exp\left(-\frac{\delta x^2 + \delta y^2}{w_0^2}\right) \cdot \exp\left(-\frac{\delta z^2}{2l_c^2}\right),$$

where l_c represents the coherence length, w_0 denotes the waist radius of the OCT beam in the sample. In our protocols, the axial (δz) and lateral (δx) motions within the B-scan plane had been compensated for using our phase-restoring subpixel image registration, and the decorrelation noise mainly originated from the out-of-plane motion (δy). Nevertheless, as shown in Supplementary Discussion Fig. 2, when the out-of-plane drift is below 2 μm, the phase uncertainty can be relatively small, with 53% of the probability being less than 0.3 rad (~20 nm in optical path length). Besides, this phase uncertainty can be further mitigated by averaging the signals across multiple pixels.

From the above analysis, we found that the impact of the out-of-plane motion on the phase uncertainty of our approach is marginal, which explains why we could extract nanoscopic phase-based ORG signals from a long-time repeated B-scan recording, even in the absence of out-of-plane motion correction. We have included these results and discussion in Supplementary Discussion 1.

References:

[Li *et al.*, 2022] H. Li, B. Tan, V. P. Pandiyan, V. A. Barathi, R. Sabesan, L. Schmetterer, T. Ling, Shot-noise limited phase-sensitive imaging of moving samples by phase-restoring subpixel motion correction in Fourier-domain optical coherence tomography. *bioRxiv*, (2022).

[Vakoc *et al.*, 2009] Vakoc, B. J., Tearney, G. J. & Bouma, B. E. Statistical properties of phase-decorrelation in phase-resolved Doppler optical coherence tomography. *IEEE Trans Med Imaging* **28**, 814-821 (2009).

[Grafe *et al.*, 2019] Grafe, M. G. O., Nadiarnykh, O. & De Boer, J. F. Optical coherence tomography velocimetry based on decorrelation estimation of phasor pair ratios (DEPPAIR). *Biomed Opt Express* **10**, 5470-5485 (2019).

Supplementary Discussion 1

Characterization of decorrelation noise associated with out-of-plane motion during a minute-long recording

To validate how phase stability can be maintained in the prolonged ORG protocol, we conducted an experiment to measure the three-dimensional eye movements over a minute-long recording. A cross-scanning pattern consisting of a horizontal scan (following the lateral direction in the repeated B-scans mode) and a vertical scan (following the out-of-plane direction in the repeated B-scans mode) was repeated over one minute. The temporal resolution was 45 ms, and the scanning areas in both directions were 12°. As demonstrated in Supplementary Discussion Figs. 1A-B, the eye movement in anesthetized rodents typically exhibited a superposition of a slow drift (<0.5 Hz) and periodic oscillations with multiple frequencies corresponding to breathing (~1 Hz) and heartbeat (~3 Hz). We separated the periodic oscillations (Supplementary Discussion Fig. 1C) and slow drift (Supplementary Discussion Fig. 1D) components by extracting their respective frequency bands (Supplementary Discussion Fig. 1B). We also calculated the residual displacements by subtracting both periodic oscillations and slow drift components from the measured displacements (Supplementary Discussion Fig. 1E).

We found that with a proper stereotaxic fixation of anesthetized rats, the peak-to-peak amplitude of the out-of-plane periodic oscillations could be reduced to within $\pm 2 \mu\text{m}$. Although we could not correct this out-of-plane movement through image registration, our previous study demonstrated that the decorrelation noise originating from a moving speckle pattern is a deterministic error rather than stochastic noise [Li *et al.*, 2022]. This suggests that the decorrelation noise, introduced by periodic oscillations at each pixel, exhibits a repetitive temporal pattern correlated with the oscillations, which can be filtered out if these oscillations stay within fixed frequency bands. Therefore, we employed bandstop filters corresponding to the frequencies and harmonics of the breathing effect and heartbeat in the subsequent data processing, which mitigated the impact of decorrelation noise caused by the periodic oscillations in the out-of-plane direction (see Supplementary Fig. 1B).

Supplementary Discussion Fig. 1. Representative three-dimensional eye movement measured in a minute-long recording. (A) The displacements along the axial, horizontal (lateral direction in the repeated B-scans mode), and vertical (out-of-plane direction in the repeated B-scans mode) directions measured using the single-step DFT algorithm [Guizar-Sicairos *et al.*, 2008]. **(B)** The amplitude spectral densities of the displacements in (A) show distinct peaks representing periodic oscillations (indicated by red bars) and slow drifts (indicated by the green bar). The periodic oscillations corresponded to breathing (~1 Hz) and heartbeat (~3 Hz). **(C)** and **(D)** are the individual periodic oscillation and slow drift components, respectively, extracted from the corresponding frequency bands in (B). **(E)** Residual displacements obtained by subtracting both periodic oscillations and drifts from the measured displacements.

On the other hand, the out-of-plane slow drift may introduce non-oscillating phase uncertainty that is more challenging to correct. Nevertheless, in our experiments, such a drift could also be maintained within $2 \mu\text{m}$ over a 1-minute recording, which was much smaller than our OCT beam size ($12.2 \mu\text{m}$, theoretical $1/e^2$ diameter, potentially even larger due to ocular aberrations). To understand the scale of decorrelation noise caused by such out-of-plane drift, we further estimated the probability density function of phase uncertainty for different out-of-plane displacements using the model proposed in OCT velocimetry [Vakoc *et al.*, 2009; Grafe *et al.*, 2019]. As shown in Supplementary Discussion Fig. 2, a lateral drift of $2 \mu\text{m}$ would result in a 53% probability of phase uncertainty being less than 0.3 rad (~20 nm in optical path length) on a single pixel, which explains the marginal impact of out-of-plane drift on our measurements. Moreover, our proposed unsupervised learning approach enables averaging the signals across multiple pixels of the same signal type to further reduce phase uncertainty during the prolonged ORG recordings.

Supplementary Discussion Fig. 2. The probability density function of the phase uncertainty resulting from out-of-plane motion (δy) [Vakoc *et al.*, 2009; Grafe *et al.*, 2019]. The theoretical diffraction-limited beam diameter at the $1/e^2$ level of the central maximum intensity was calculated to be $12.2 \mu\text{m}$ in our OCT imaging system.

Line 111: “Regarding the uncorrected out-of-plane motion, our experiments demonstrated that even for one-minute-long recordings, the out-of-plane slow drift was typically below $2 \mu\text{m}$ (see Supplementary Discussion 1 and Supplementary Discussion Fig. 1) - much smaller than the beam diameter ($12.2 \mu\text{m}$, theoretical $1/e^2$ width, potentially even larger due to ocular aberrations). The phase uncertainty resulting from such uncorrected out-of-plane slow drift remained marginal, which allowed reliable measurements of retinal dynamics throughout prolonged ORG recordings (see Supplementary Discussion 1 and Supplementary Discussion Fig. 2).”

Line 320: “The theoretical diffraction-limited beam diameter was $12.2 \mu\text{m}$ ($1/e^2$ width) with a standard rat eye model.”

(1.2) The author used machine learning for the signal and signal source analysis. It was a good attempt, however, it needs to be reconsidered. In this study, the author adopted unsupervised machine learning for the classification of two kinds of signals in the retinal band including OS tips and RPE cells. I have some questions and concerns: (1) In Figure 1A the vis-OCT B-scan we can see that the “mixed” band contains only the OS tip and the RPE layers. Why did the author intentionally separate it into four layers instead of two? This separation sequentially gave the complicated results in Figure 1D. Additionally, because separation into four, each band was very thin, which inevitably caused high fluctuation and segmentation errors.

[Reply] As the mixed band contains only the OS tip and the RPE layers, we did try to separate it into two layers. However, due to the limited axial resolution of OCT and the interdigitation between the OS tip and RPE, we couldn’t identify a clear boundary within the mixed band that would result in distinct tissue dynamics on opposite sides. Instead, we observed a gradual change in tissue dynamics when moving the region of interest for spatial averaging from the very top (dark red) to the very bottom (dark blue), as illustrated in Fig. 1D. The purpose of Fig. 1D was to highlight such gradual change in tissue dynamics within the mixed band, and we used four layers instead of two to provide sufficient granularity for depicting this gradual change.

(1.3) In the spatial-temporal feature space, the author used two of the three clusters to represent the two types of ORG, how about the third type, it is just a transition of the other two types? Where is type III located in the “mixed” band?

[Reply] In the revised manuscript, we present the distributions of all three signal types on each two-dimensional feature plane and along individual feature dimensions in Supplementary Fig. 3. Since all signals are distributed along a continuous arc in the spatiotemporal feature space (Supplementary Fig. 3) and Type-III signals consistently fall between the Type-I and Type-II signals, we consider Type-III as a transition band between the other two types. The locations of Type-III signals along the spatial feature (depth) are plotted in Supplementary Fig. 3D.

Supplementary Fig. 3. (A) The distributions of three signal types in the spatiotemporal feature space (red: Type-I, blue: Type-II, brown: Type-III), and (B)-(D) their projections onto 2D feature planes. The 1D histograms in (B) and (D) show the distributions of three signal types along each feature.

Line 162: “By thresholding the dendrogram at the solid black line in Fig. 2B, the remaining phase traces (pink dots in Fig. 2A) were grouped into three clusters in the spatiotemporal feature space (Fig. 2C). A transition band (brown dots) facilitated better separation between the two distinct light-evoked tissue dynamics (Supplementary Fig. 3).”

(1.4) From Figure 3B, we can see that the type I signal correlates with the OS tip, and the type II signal correlates with the RPE. Does it mean we can conclude that the signal type is mostly determined by the spatial information (depth)? If this is true, we get back to the point that the two kinds of signals are separated by the layer segmentation (depth difference).

[Reply] As depicted in Supplementary Fig. 3D (see above), there is indeed a partial correlation between signal types and depth. However, because Type-I and Type-II signals overlap with Type-III signals (see the histogram on the right in Supplementary Fig. 3D) over an extensive depth range, we cannot separate Type-I and Type-II signals by solely referring to the depth difference. Temporal features also play a significant role in our signal classification, as shown in Supplementary Fig. 3B.

Minor comments:

(1.5) The author may consider discussing the relationship between this study and other studies. Multilayer ORG and longtime monitoring [2], mouse phase ORG, also longtime monitoring [3].

References

- [2] C. D. Lu, B. Lee, J. Schottenhamml, A. Maier, E. N. Pugh, and J. G. Fujimoto, "Photoreceptor layer thickness changes during dark adaptation observed with ultrahigh-resolution optical coherence tomography," *Investigative Ophthalmology Visual Science* 58, 4632-4643 (2017).
- [3] E. Pijewska, P. Zhang, M. Meina, R. K. Meleppat, M. Szkulmowski, and R. J. Zawadzki, "Extraction of phase-based optoretinograms (ORG) from serial B-scans acquired over tens of seconds by mouse retinal raster scanning OCT system," 12, 7849-7871 (2021).

[Reply] We have revised the Discussion section to include these studies.

Line 283: "Several studies investigated prolonged ORG signals using different methodologies. Lu *et al.* studied the changes in retinal layer thickness after strong visual stimuli (>23% rhodopsin bleach) by segmenting the layers from OCT structural images [Lu *et al.*, 2017]. They examined micrometer-level responses during a 30-minute dark adaptation process after the visual stimulus, with a temporal resolution of 2.1 seconds and a baseline signal fluctuation of hundreds of nanometers. Zhang *et al.* investigated the OCT intensity profile change in depths by averaging numerous A-scans. They observed a slow (peak latency of 10-100 seconds, depending on the bleach level) increase in distance between IS/OS and BrM in response to light, and attributed it to the elongation of the OS [Zhang *et al.*, 2017]. Notably, this slow signal resembles our SRS expansion, while the actual OS signal in our observations and previous reports [Azimipour *et al.*, 2020] rises and recovers much faster. A follow-up study by Pijewska *et al.* demonstrated that the phase-based ORG signals enabled higher detection sensitivity than the intensity-based processing methods [Pijewska *et al.*, 2021]. Their results, obtained using strong stimuli (100% rhodopsin bleach), showed almost linear expansions between the ELM and BrM over 40 seconds after the visual stimuli were delivered."

(1.6) The author used the vis-OCT to show retinal layers that were not shown in the NIR-OCT B-scans. Later, the author also used the vis-OCT B-scan outer bands for the signal source investigation. In general, the vis-OCT has a better resolution, which should show more details than the NIR-OCT. However, in this study, the authors need to pay attention to the fact that the retinal outer bands can be changed by dark or light adaptation conditions [4]. Normally the retina being imaged by vis-OCT should be light-adapted, and if the NIR-OCT image in the figure was from a dark-

adapted retina, the comparison is not fair. In other words, to precisely guide the NIR-OCT outer band using a vis-OCT B-scan is risky, unless the adaptation condition is justified.

Reference

[4] T.-H. Kim, J. Ding, and X. Yao, "Intrinsic signal optoretinography of dark adaptation kinetics," *Scientific reports* 12, 2475 (2022).

[Reply] We thank the reviewer for raising this question regarding possible outer band change due to the light adaptation in vis-OCT. To investigate this matter further, we conducted an additional experiment. We dark-adapted a Brown Norway rat for 12 hours and used the same NIR-OCT system to image the retina in a dark room – the same dark adaptation condition as in the ORG experiment. A volumetric scan (500 A-scans \times 125 B-scans) was collected within a $12^\circ \times 0.3^\circ$ (0.8 mm \times 0.02 mm) rectangular field of view. The camera speed was set to 100 kHz to enhance the signal-to-noise ratio. Neighboring B-scans were aligned and subsequently averaged into a single frame to reduce image speckles (see Fig. 1B below). Since we can resolve the RPE layer from this new despeckled NIR-OCT B-scan image, the content related to Vis-OCT has been removed from the revised manuscript to avoid potential confusion.

Fig 1. Retinal layers and their dynamics in response to visual stimuli. (A) Averaged retinal B-scan ($n = 125$) from the same location. **(B)** Despeckled retinal B-scan, generated by averaging individual B-scans ($n = 125$) from a neighboring region, allows better resolving photoreceptors' OS, RPE, and BrM layers. An enlarged view of the magenta dashed box is shown in Fig. 3B. Both scans were acquired after 12 hours of dark adaptation. Scale bar: 50 μm .

Accordingly, we used the intensity profile of the outer retina bands in the despeckled NIR-OCT B-scan image to verify the ORG signal origin, where the location of the Type-I signal (red bins) corresponded to the intensity profile of OS, and the location of the Type-II signal (blue bins) corresponded to the location of RPE, respectively. We have incorporated this information in the updated Fig. 3B.

Fig. 3B: An enlarged view of the dashed box in Fig. 1B, with contrast adjustment to enhance the RPE visibility. The histograms of Type-I (red bins) and Type-II (blue bins) signals, fitted by Gaussian functions (solid lines), display the depth distribution of the signals overlaid on top of the despeckled structural image, with a yellow line representing the averaged intensity profile. The colored bars on the left correspond to the depth range from which the signals in Fig. 1D were extracted.

Line 99: “A despeckled image, generated by averaging the neighboring B-scans from a rectangular region, was used to validate the delineation of retinal layers (Fig. 1B). As illustrated in Fig. 1, several hyperreflective layers were observed in the outer retina, including the external limiting membrane (ELM), the inner segment/outer segment junction (IS/OS), Bruch’s membrane (BrM), and a thick speckling layer between the IS/OS and the BrM, which we call the *mixed layer* for convenience. As confirmed by the despeckled image, the mixed layer comprised photoreceptors’ OS and RPE cells.”

Line 181: “The locations of Type-I and Type-II signals corresponded to the OS and RPE in the intensity profile, respectively, as determined by the despeckled image (yellow line in Fig. 3B).”

Line 334: “To generate a despeckled retinal image, volumetric raster scans (500 A-scans × 125 B-scans) were collected within a 12° × 0.3° (0.8 mm × 0.02 mm) rectangular field of view. The camera speed was set to 100 kHz to enhance the signal-to-noise ratio. Neighboring B-scans were aligned and then averaged into a single frame [Kho *et al.*, 2020].”

[Kho *et al.*, 2020] Kho, A. M., Zhang, T., Zhu, J., Merkle, C. W. & Srinivasan, V. J. Incoherent excess noise spectrally encodes broadband light sources. *Light: Science & Applications* 9, 172 (2020).

(1.7) In line 84, “a 12° field of view (FOV)”, the author may consider adding the actual length like “xxx um” to help the reader to understand it better. We can see there is information in the material and method part, however, it is not very easy for the readers to find it.

[Reply] We thank the reviewer for the suggestion. The 12° field of view in our acquisition corresponds to 0.8 mm [Wang *et al.*, 2022]. We added this information to the revised manuscript.

[Wang *et al.*, 2022] Wang, B.-Y. *et al.* Electronic photoreceptors enable prosthetic visual acuity matching the natural resolution in rats. *Nature Communications* **13**, 6627 (2022).

Line 98: “Cross-sectional (Fig. 1A) and volumetric scans were acquired in time sequences from a 12° field of view (FOV), corresponding to 0.8 mm on wild-type rat retinas [Wang *et al.*, 2022].”

(1.8) In Figure 1. For the NIR and vis-OCT B-scans, it seems like they are not from the same processing steps. The vis-OCT B-scan seems like is averaged by more frames. To show a better comparison, the author may consider processing the two images by the same method.

[Reply] As stated in (2), we replaced the vis-OCT image with a despeckled OCT image acquired with the same NIR-OCT machine used for the ORG experiment. Following the reviewer’s suggestion, we averaged same number of registered individual B-scans (n = 125) to generate Fig. 1A and Fig. 1B. Notably, obvious speckle patterns can be seen in Fig. 1A, and the speckle patterns were reduced in Fig. 1B. Such difference was caused by the different scanning protocols: Fig. 1A averaged individual B-scans acquired from the same location, while Fig. 1B averaged neighboring B-scans acquired from a rectangular region (0.8 mm × 0.02 mm). Both images were cropped in the lateral dimension to a field of view of 0.2 mm.

(1.9) In line 96, the citation is “bioRxiv (2022)”, if it is officially published, please update the information.

[Reply] Our work on phase-restoring subpixel motion correction is currently under revision for the *IEEE Transactions on Medical Imaging* journal, with a somewhat lengthy review process. We will link the bioRxiv preprint to the published version when it is available.

(1.10) In line 97, the author said “Note that in this study, when converting the phase change into the OPL change, an additional negative sign was added if the target layer was anterior to the reference layer. This ensured that an increase in OPL always represented an expansion between the two layers.” This sentence is confusing. This sentence is confusing and may lead to a possible interpretation that “OPL change” is always positive. the author may consider saying something like “The relative phase between two layers is calculated by subtracting the upper layer from the deeper layer, to ensure the relative phase is always positive”.

[Reply] We thank the reviewer for pointing out this confusion. We have refined the expression to avoid possible ambiguity.

Line 119: “Note that in this study, the temporal phase change was extracted from a target layer with respect to a reference layer, which could be located either anterior or posterior to the target layer (see Supplementary Method 1). For consistency in interpretation of the increase/decrease of OPL as the expansion/contraction between the two layers, a negative sign was added when converting the phase change into the OPL change if the target layer was anterior to the reference layer.”

Supplementary Method 1:

“This ensured that an increase/decrease in OPL change (Δ OPL) consistently represented an expansion/contraction between the target layer and the reference layer.”

(1.11) In line 166, the author explains the background illumination for the light adaptation. Can the author explain more about what the bleaching level these illumination backgrounds can cause?

[Reply] If the regeneration of rhodopsins is neglected, the 5-minute background illumination at 6×10^4 photons/ $(\mu\text{m}^2 \cdot \text{s})$ would bleach 20.3% rhodopsins. We added this information to the caption of Fig. 4.

Figure 4, caption

“If the regeneration of rhodopsins is neglected, the 5-minute background illumination at 6×10^4 photons/ $(\mu\text{m}^2 \cdot \text{s})$ would bleach 20.3% rhodopsins.”

Reviewer #2:

(2.1) Given that electroretinography permits direct (en masse) recording of neuronal electrical responses (which may not precisely correlate with optical/osmotic changes) including from inner retinal neurons, I would not regard ORG as necessarily a replacement for electroretinography, but providing complementary information, likely with higher spatial resolution, particularly physiological data not directly obtainable with ERG.

[Reply] We have rephrased our statement accordingly.

Line 43: “This new technique expands the diagnostic capabilities and practical applicability of optoretinography, providing an alternative to electroretinography, while combining structural and functional retinal imaging in the same OCT machine.”

Line 231: “This new technique expands the practical applicability of optoretinography to studies of not only photoreceptors but also the RPE’s control of water dynamics in SRS in health and disease, thus providing an alternative to ERG.”

(2.2) Line 101: “distance between ELM and BrM increases after the stimulus (1 ms, 500 nm, 0.18% bleach level).”

The 500 nm here is slightly confusing as to whether it refers to the distance or stimulus wavelength. I would suggest amending to, “distance between ELM and BrM increases after the stimulus (1 ms, wavelength 500 nm, 0.18% bleach level).”

[Reply] We thank the reviewer for pointing out this ambiguity. The sentence has been revised accordingly.

Line 126: “distance between ELM and BrM increases after the stimulus (1 ms, wavelength 500 nm, 0.18% bleach level).”

(2.3) Line 123: “we combined phase traces from five rats for feature extraction”

How were these 5 rats chosen? How many males/females? From the Methods, 43 rats were used in total with Table S2 giving the numbers in each test.

[Reply]: These 5 rats (3 males and 2 females, 12 weeks old) were a separate group apart from the 43 rats listed in Table S2. We first randomly selected 5 rats to establish a feature space and pre-trained a support vector machine (SVM). The phase traces extracted from all 43 rats listed in previous Table S2 were later projected onto the same feature space and classified consistently using the pre-trained SVM. We have updated Supplementary Table 1 and Supplementary Table 2 to incorporate this information.

Line 153: “we combined phase traces from five rats for feature extraction and subsequent unsupervised clustering (Supplementary Table 1 and Supplementary Table 2).”

Line 341: “Brown Norway rats (N = 48) were used with details listed in Supplementary Table 1.”

(2.4) Line 153: “By incorporating the Type-II signal with the dynamics between IS/OS and ELM, we obtained the dynamics of SRS (from ELM to RPE).”

It was not clear to me precisely how the dynamics of the subretinal space were derived. Could the authors explain this more clearly?

[Reply] The SRS is defined as the extracellular space that surrounds photoreceptors, and it spans from the apical membrane of the RPE to the ELM [Lindell *et al.*, 2023]. The dynamics between the IS/OS and ELM can be calculated by averaging signals extracted from pixels in the ELM layer, using the IS/OS layer as a reference. By summing up the Type-II signal (from RPE to IS/OS) and the OPL change between IS/OS and ELM, we can directly obtain the SRS dynamics. We elaborated on this procedure in the revised manuscript.

Reference:

[Lindell *et al.*, 2023] Lindell, M. *et al.* Volumetric Reconstruction of a Human Retinal Pigment Epithelial Cell Reveals Specialized Membranes and Polarized Distribution of Organelles. *Invest. Ophthalmol. Vis. Sci.* **64**, 35-35 (2023).

Line 189: “We further investigated the dynamics of the SRS, the extracellular space surrounding photoreceptors and spanning from the apical membrane of RPE to the ELM [Lindell *et al.*, 2023]. The dynamics between IS/OS and ELM can be calculated by averaging the signals extracted from the pixels in the ELM layer, using the IS/OS layer as a reference. By summing up the Type-II signal (distance from RPE to IS/OS) and the OPL change between IS/OS and ELM, we obtained the SRS dynamics.”

(2.5) Line 318: “A low-pass filter with a cut-off frequency of 10 Hz was subsequently used to filter out high-frequency oscillations”

Given that much of phototransduction occurs within 100 ms, is there a possibility that physiologically meaningful data were lost/distorted by application of this filter? Did/can the authors check the effect of different cut-off frequencies and provide evidence or rationale to support this cut-off?

[Reply] We thank the reviewer for raising this question. Indeed, applying linear filters to transient signals may lead to loss or distortion of physiologically meaningful signals if the signal processing pipeline is not properly designed. As the reviewer suggested, we have confirmed that the filtered Type-I and Type-II signals did not exhibit significant differences compared with the unfiltered signals (see the figure below). Furthermore, we only used the low-pass filter to classify the two types of ORG signals (Fig. 2). After the pixels in the mixed layer had been classified, we didn't apply

the low-pass filter to the final light-evoked dynamics of the retinal layers. Following this approach, the low-pass filter should have minimum impact on the optical signatures of the light-evoked retinal dynamics that we extracted from the ORG recordings. In the revised manuscript, we updated Fig. 2D to show the Type-I and Type-II signals obtained without the low-pass filter applied to prevent potential confusion. Meanwhile, we emphasized that the low-pass filter was only implemented for the signal classification in the corresponding section of Methods.

Averaged OS and SRS signals before and after filtering.

Fig. 2D:

Line 394: “For individual phase traces, our filters effectively captured the signal profiles while suppressing both periodic oscillations and high-frequency noise (Supplementary Fig. 1B). Meanwhile, it should be noted that the low-pass filter was implemented exclusively for the signal classification. After identifying pixels corresponding to the Type-I and Type-II signals, the light-evoked responses reported in this article were not processed by any low-pass filter to prevent potential signal distortion.”